# Radiomics Initialized Deep Embedding Network (RIDE-Net) to Prognosticate Survival in Renal Cancers

Brennan Flannery
*Department of Biomedical Engineering*
*Case Western Reserve University*
Cleveland, OH, United States of America
bxf169@case.edu

Thomas DeSilvio
*Department of Biomedical Engineering*
*Case Western Reserve University)*
Cleveland, OH, United States of America
tgd15@case.edu

Satish E. Viswanath, PhD
*Department of Biomedical Engineering and Pediatrics*
*Emory University*
Atlanta, GA, United States of America
satish.viswanath@emory.edu

*Abstract*—Popularly used machine learning approaches for medical imaging offer complementary advantages for disease characterization. Hand-crafted computational features (or radiomic features) offers generalizable quantitative measures which can be intricately linked to disease phenotypes, while iteratively optimized convolutional neural networks (CNNs) offer complex features with robust performance to imaging variations. To address the question of how best to exploit the relative advantages of both approaches, we introduce Radiomics Initialized Deep Embedding Network (RIDE-Net) which leverages intuitive radiomic descriptors to enhance the performance of a CNN model. Our approach involves: (1) identifying disease-specific radiomics features associated with an end-point of interest, (2) pre-training a residual learning network to directly predict these specialized radiomic features, and (3) optimizing this primed RIDE-Net to predict the outcome of interest. We evaluate RIDE-Net in the context of prognosticating overall survival for renal cancers using a multi-institutional cohort of 510 patients and a pre-training cohort of 5195 patients with CT volumes. A RIDE-Net based survival model achieved c-indices of 0.67 and 0.66 in testing and holdout validation, respectively, which significantly outperformed a standard radiomics based Cox regression model (0.56, 0.62) as well as a standard ResNet based survival model (0.60, 0.62) with significantly less variation across training runs. We also found that RIDE-Net deep features achieve increased reproducibility compared to standard radiomic features in terms of intra-class correlation coefficient and can achieve similar prognostic performance even when utilizing 5% of the training data (c-index of 0.66 in validation via few shot analysis). RIDE-Net thus represents a novel integrated approach to combining the strengths of radiomics and deep learning for robust, accurate, and reproducible predictions in medical imaging tasks.

*Index Terms*—deep learning, radiomics, computed tomography

Research reported in this publication was supported by the National Cancer Institute (1R01CA280981-01A1), the National Institute of Nursing Research (1R01NR019585-01A1), the National Heart, Lung, and Blood Institute (1R01HL165218-01A1), the National Science Foundation (Award 2320952), the Veterans Affairs Biomedical Laboratory Research and Development Service (1I01BX006439-01), the DOD Peer Reviewed Cancer Research Program (W81XWH-21-1-0725), the Ohio Third Frontier Technology Validation Fund, the JobsOhio Program, NIBIB (T32EB007509), and the Wallace H. Coulter Foundation Program in the Department of Biomedical Engineering at Case Western Reserve University. This work made use of the High Performance Computing Resource in the Core Facility for Advanced Research Computing at Case Western Reserve University.

The content is solely the responsibility of the authors and does not necessarily represent the official views of the National Institutes of Health, the U.S. Department of Veterans Affairs, the Department of Defense, or the United States Government.

## I. INTRODUCTION

Recent advances in machine learning (ML) have leveraged hand-crafted computational intensity, shape, or texture features (also known as radiomics) as well as convolutional neural networks (CNNs) toward improved disease classification [1], [2], response prediction [3], as well as prognosticating survival [4], [5]. When provided with sufficient training data, CNNs have achieved high inference accuracy, often surpassing traditional machine learning methods [6] based on extracting disease-related features without manual specification. With the use of image augmentations during training, CNNs demonstrate resilience to input differences based on scanners, institutions, or other batch effects [7]. By contrast, well-designed and specialized radiomic features have achieved generalizable performance even when utilizing limited data cohorts [8], as well as demonstrating associations with specific biological phenotypes [2]. The contrasting approaches by which radiomic descriptors and deep learned features are optimized suggest they may be inherently complementary to each other. In fact, only weak to moderate correlation has been observed between radiomic and deep features in previous empirical analyses [5], [9], indicating these approaches may indeed quantify distinct information with regard to disease characteristics. This in turn suggests an intelligent approach to exploiting and integrating

these techniques could yield more generalizable, accurate, and reproducible ML models.

## A. Previous Work and Novel Contributions

Broadly speaking, previous work on integrating radiomics and convolutional neural network (CNN) approaches can be categorized as *feature-level* or *decision-level* approaches. Feature-level fusion involves concatenating or merging radiomics and deep features into a unified representation prior to feeding them into a separate classification or prognostication DL network [3], [6], [10]–[14]. By contrast, decision-level fusion involves training separate radiomics and deep learning models, followed by ensembling their predictions to enable improved classification or prognostication [15], [16]. An alternative approach has also been to simply utilize deep CNN features as inputs for traditional ML models such as support vector machines (SVMs) or decision trees [17]. Unfortunately, such decision-level fusion may be prone to overfitting, in addition to not explicitly accounting for complementary information between radiomics and deep features, thus limiting the predictive power of the ensembled model.

While feature-level fusion partially addresses this limitation by enabling individual feature types to be weighted, it may actually amplify weaknesses of the individual methodologies. For instance, CNN performance is known to be highly dependent on the composition of the training dataset, typically requiring large amounts of balanced data to achieve robust generalization [8], [17]. Similarly, multiple radiomics studies have suggested reproducibility challenges [8], [18], [19] due to variations in scanner manufacturers, acquisition parameters, and imaging protocols across data cohorts. In other words, integrating these methods will need to exploit the interpretable aspects of radiomics features (i.e., which capture generalizable biological associations) while also leveraging the reliability and robustness of CNNs to imaging variations and batch effects (through optimization over image augmentations).

## B. Study Goal

In this work, we propose a novel Radiomics Intialized Deep Embedding Network (RIDE-Net), which exploits radiomic descriptors as a pre-training objective for a residual learning network, followed by later optimization for robust and accurate performance in downstream tasks. The RIDE-Net architecture involves first priming a ResNet model to predict specialized radiomic descriptors that have been selected for their association with a specific end-point, thus yielding a deep embedding that is intricately linked to the target disease. We evaluated the predictive power of RIDE-Net deep features in the context of prognosticating overall survival of renal cancers via diagnostic CT scans, with multi-institutional validation (N=520 patients, 10 institutions). In addition to comprehensively comparing RIDE-Net performance to radiomic and CNN model performance individually, we also examine few-shot performance compared to a standard ResNet, as well as a detailed analyses of RIDE-Net feature reproducibility and model variability.

## II. METHODS

RIDE-Net encodes domain knowledge via radiomic features into neural network weights using an integrated and label-agnostic pre-training process. This is accomplished through three distinct steps:

**Step 1. Radiomics Initialization**: A subset of $M$ radiomic features, denoted $F_i^R, i \in \{1, \ldots, M\}$, are first selected based on their association with a specific outcome of interest, denoted $Y$ (e.g. cancer grade or stage, time to an event). Feature selection is typically implemented via methods such as LASSO [20] or mRMR to distill a much larger feature space to a subset of outcome-specific radiomic descriptors (Fig. 1A).

**Step 2. RIDE-Net Training**: A separate unlabeled pre-training cohort without outcome information is utilized to train RIDE-Net to output a corresponding set of $M$ deep features, denoted $F_i^D, i \in \{1, \ldots, M\}$. Pre-training allows for model refinement from a diverse pool of data with extensive imaging variability, thus encoding information contained in $F^R$ into RIDE-Net. Robustness of $F^D$ to input variability can be enhanced by leveraging image augmentations during pre-training. RIDE-Net is optimized over multiple epochs, iteratively embedding outcome-specific radiomic descriptors into DL model weights based on the loss function: $\mathcal{L}_{MSE} = \frac{1}{M} \sum_{i=1}^{M} \left( F_i^R - F_i^D \right)^2$. (Fig. 1B)

**Step 3. RIDE-Net Optimization**: A linear layer is added to the pre-trained RIDE-Net, which is optimized for predicting the outcome $Y$ directly, thus adapting the radiomics-based model weights within RIDE-Net for the desired task. Using only a single additional layer provides flexibility for optimization without susceptibility to overfitting. Optimization is implemented using a task appropriate loss function (e.g. mean squared error for regression, binary cross entropy for classification, Cox loss for survival) (Fig. 1C).

## III. EXPERIMENTAL DESIGN

**Data Description**: RIDE-Net was evaluated for the task of prognosticating overall survival for renal cancers in a multi-institutional setting, utilizing:

1) **Survival cohort (denoted $C^s$)** included contrast CT scans curated from the publicly available KITS19 (N=210), Kidney Renal Clear Cell Carcinoma (KIRC, N=267), Kidney Chromophobe (KICH, N=15), and Kidney Renal Papillary Cell Carcinoma (KIRP, N=33) collections from The Cancer Imaging Archive. All patients had a confirmed diagnosis of renal cancer, together with last followup time and death status (30% of the patients experienced death events). $C^s$ encompassed diverse imaging protocols and kidney cancer phenotypes (primarily clear cell renal carcinoma) from 10 institutions. The survival dataset was split into 60% training ($C_{tr}^s$), 20% testing ($C_{te}^s$), and 20% holdout validation ($C_{hv}^s$).

2) **Pre-training cohort (denoted $C^p$)** comprised CT scans from the public Abdomen Atlas collection (N=5195) [21] comprising a variety of contrast enhancements

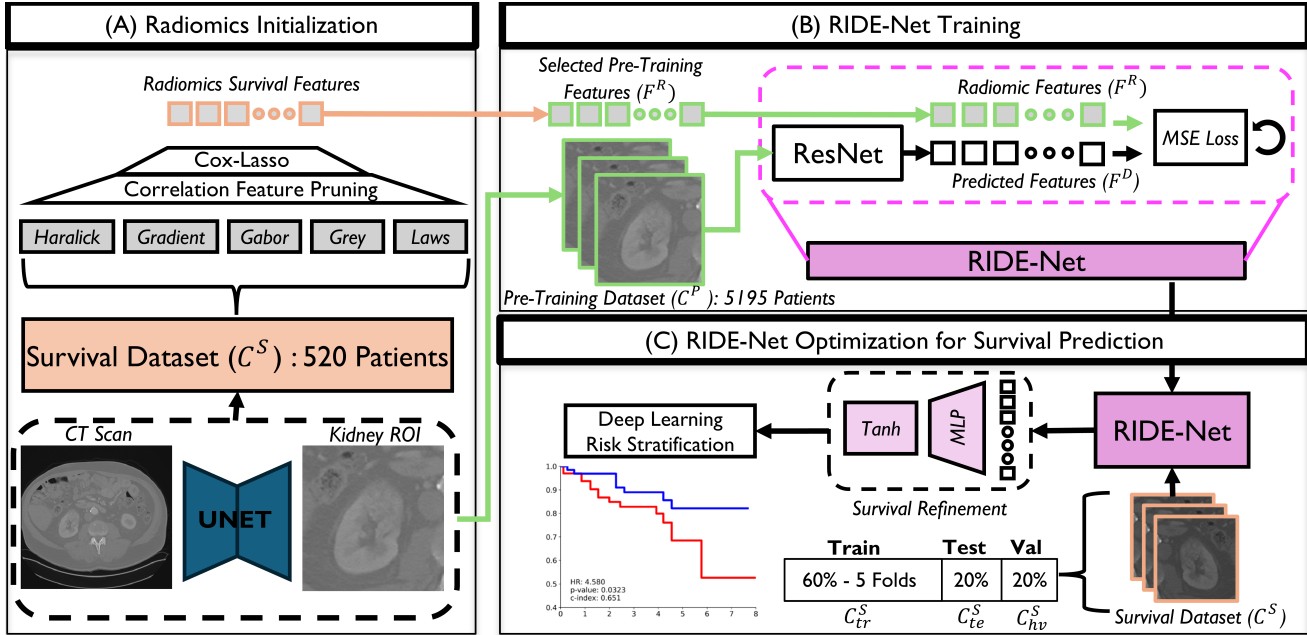

Fig. 1. Overall methodological workflow for constructing and optimizing RIDE-Net. (A) Radiomics features are chosen from a pool of 1640 features from 5 feature families measuring diverse aspects of radiological imaging presentation. Correlated features are removed before LASSO selects the defined number of pre-training features. (B) The selected radiomics features are used as direct pre-training targets on a large unlabled cohort to prime a Resnet18 network for downstream survival prediction. (C) A single trainable MLP and activation function was added to the network to refine pre-trained features for survival prediction.

and acquisition times, accrued from 76 institutions. No outcome or disease status information was available for these patients. The pre-training dataset was randomly split into 80% training and 20% testing.

All CT scans underwent trilinear interpolation to 1mm x 1mm x 1mm resolution. A U-net was trained for kidney segmentation and used to identify a 100 x 100 x 100 pixel ROI centered on each kidney for every patient. For $C^s$, only the kidney with the tumor lesion was included, while both kidneys were included in $C^p$.

**Radiomics Initialization**: A total of 1640 3D radiomic features were extracted from each kidney ROI, based on varying parameters associated with radiomic operators from the Haralick [22], gradient [23], Gabor [24], statistical, and Laws [25] feature families. The mean value of each feature was computed within each ROI, followed by feature normalization to ensure that all radiomic features lay within a comparable range (mean 0, std 1). Next, highly correlated features were removed to yield 137 "uncorrelated" radiomic features; followed by selecting the $M$ most relevant radiomic features (corresponding to $F_i^R, i \in \{1, \ldots, M\}$) associated with survival outcomes in $C_{tr}^s$ using a Cox proportional hazards model ($\mathcal{H}^{RM}$) with LASSO regression. Performance was quantified via $C_{te}^s$ and $C_{hv}^s$.

**RIDE-Net Training**: Radiomic features $F_i^R, i \in \{1, \ldots, M\}$, were first extracted from the kidney ROI for every scan in $C^p$. A 3D ResNet-18 was trained to output $F_i^D, i \in \{1, \ldots, M\}$, based on optimizing $\mathcal{L}_{MSE}$ over 200 epochs with a batch size of 8, learning rate of 0.001, and a

weight decay of 0.0001 with an Adam optimizer. The model was evaluated on the testing set in $C^p$ after each epoch, and the model with the lowest test loss was retained. The final result was the pre-trained RIDE-net model, encoding survival-specific radiomic features.

**RIDE-Net Optimization and Survival Prediction**: The additional layer and hyperbolic tangent activation function added to RIDE-Net utilizes $F^D$ to output a survival prediction. The resulting model, denoted $\mathcal{H}^S$, outputs a deep risk score which indicates a patient's risk for death. $\mathcal{H}^S$ was optimized through 5-fold cross validation within $C_{tr}^s$, while using a Cox proportional hazards inspired loss function over the course of 200 epochs, a batch size of 16, learning rate of 0.001, learning rate decay of 0.0001, and early stopping with an Adam optimizer. $\mathcal{H}^S$ was evaluated after each epoch on the hold out test fold as well as on $C_{te}^s$, and the model with the lowest test loss was saved. The best model across all 5 folds was selected as the final optimized $\mathcal{H}^S$, which was evaluated on $C_{hv}^s$. For comparison, a 3D Resnet-18 (denoted $\mathcal{H}^{R18}$) was implemented to utilize the kidney ROI on each CT scan as an input for predicting survival which was optimized via the same train/test setup within $C^s$.

**Model evaluation**: $\mathcal{H}^{RM}$, $\mathcal{H}^S$, and $\mathcal{H}^{R18}$ were compared in terms of concordance indices (c-index). One-way ANOVA with post-hoc Tukey tests was used to compare performance across different models, with p-values adjusted for multiple comparisons using the Holm-Bonferroni correction ($\alpha = 0.05$). To compare the variability of survival prediction accuracy, both $\mathcal{H}^S$ and $\mathcal{H}^{R18}$ were re-optimized with 50 differ-

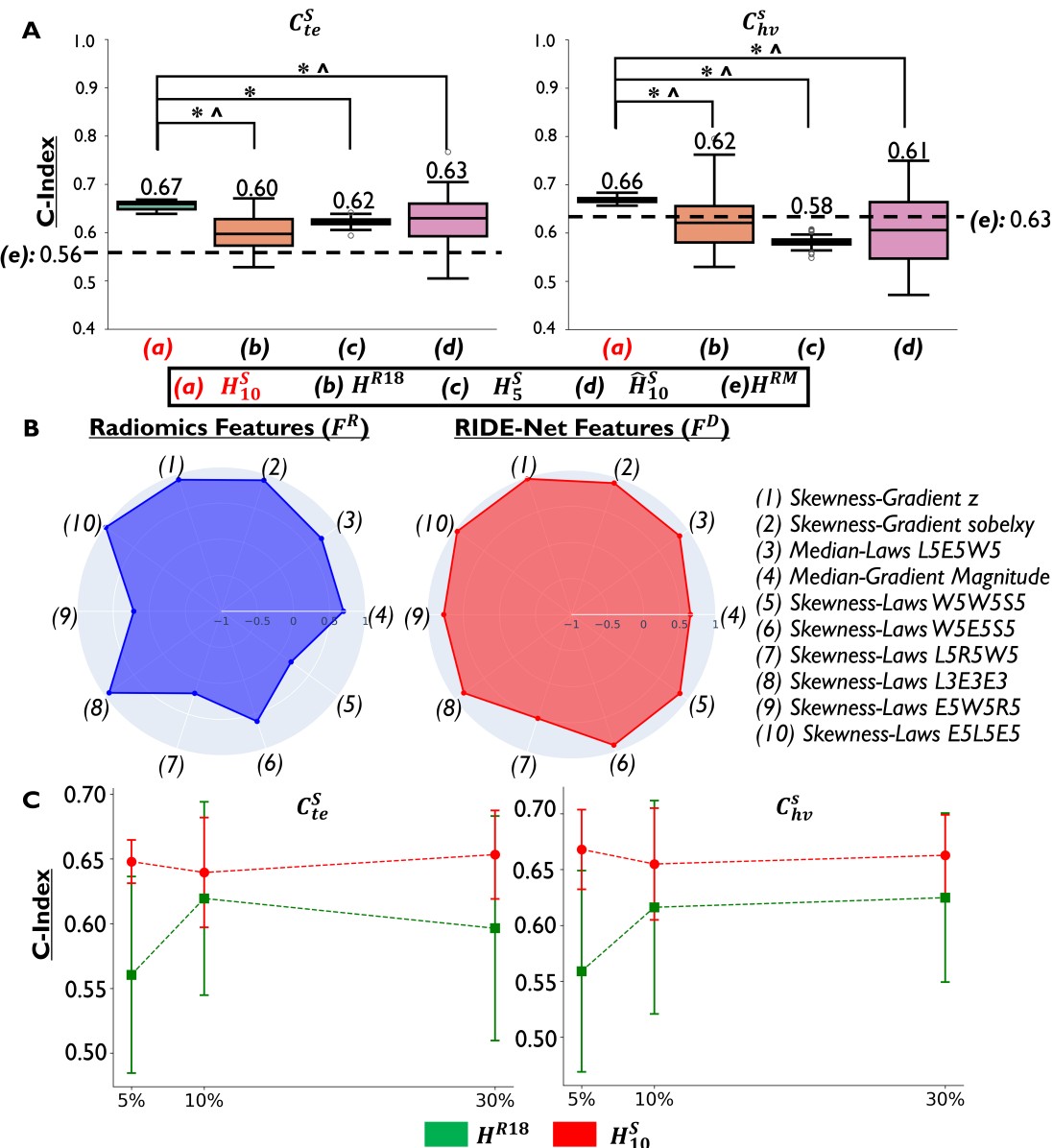

Fig. 2. (A) Box plots of c-index across 50 different model initializations and optimizations for each of $\mathcal{H}_{10}^S$, $\mathcal{H}^{R18}$, $\mathcal{H}_5^S$, and $\widehat{\mathcal{H}_{10}^S}$, where $*$ indicates the means of the indicated distributions are significantly different ($\alpha < 0.05$ in one-way ANOVA and post-hoc Tukey testing) while $\wedge$ indicates the variances of the distributions are significantly different ($\alpha < 0.05$ in pairwise F-testing). (B) Radial graphs comparing ICC values to examine reproducibility of radiomic features $F_i^R$ and deep features $F_i^D$, $i \in \{1, \ldots, M\}$, to small image perturbations . (C) Few-shot analysis comparing c-index performance of $\mathcal{H}^S$ and $\mathcal{H}^{R18}$ when considered only 5%, 10%, or 30% of $C_{tr}^s$ during model optimization

ent random model weight initializations. Variance differences between model configurations were assessed using pairwise F-tests ($\alpha = 0.05$), with corrected p-values. Reproducibility of $\mathcal{H}^S$ features were compared against $\mathcal{H}^{RM}$ features by first inducing small perturbations in the kidney ROI through random image augmentation (using a subset of patients from $C_p$), followed by evaluating feature reproducibility for each feature set based on interclass-correlation coefficient (ICC). Few-shot analysis for $\mathcal{H}^S$ and $\mathcal{H}^{R18}$ were conducted by limiting $C_{tr}^s$ to 5, 10, or 30% of the total dataset when training each model. To evaluate how the number of radiomic features

used in pre-training RIDE-Net impacted downstream survival prediction performance, two different RIDE-Net models, $\mathcal{H}_5^S$ ($M = 5$) and $\mathcal{H}_{10}^S$ ($M = 10$), were constructed and evaluated. Different approaches to RIDE-Net optimization were examined, by comparing $\mathcal{H}_{10}^S$, where only the added linear layer was trainable, to $\widehat{\mathcal{H}_{10}^S}$, where every layer in RIDE-Net as well as the added linear layer was trainable.

## IV. RESULTS AND DISCUSSION

The RIDE-Net based survival model, $\mathcal{H}_{10}^S$, achieved the best overall c-index of 0.66 on the $C_{te}^s$ and 0.67 on $C_{hv}^s$, repre-

senting a significant improvement over alternative strategies (Fig. 2A: c-index of $\mathcal{H}^{R18}$ is 0.60, 0,62 and c-index of $\mathcal{H}^{RM}$ is 0.56, 0.62). This indicates that RIDE-Net can consistently outperform both standard deep learning and radiomics-based approaches for prognosticating overall survival in renal cancers.

When comparing $\mathcal{H}_{10}^{S}$ to $\widehat{\mathcal{H}_{10}^{S}}$, the latter achieved significantly lower c-indices of 0.62 and 0.63 on $C_{te}^{s}$ and $C_{hv}^{s}$, respectively. This suggests that freezing RIDE-Net weights during downstream refinement serves as an effective initialization strategy, preserving critical domain knowledge encoded during pre-training while allowing for targeted fine-tuning on the task-specific dataset.

When the number of radiomic features were varied, $\mathcal{H}_{5}^{S}$ achieved significantly lower c-indices of 0.62 and 0.57 in testing and validation, respectively, indicating that the performance of RIDE-Net is sensitive to the number of radiomic features used during pre-training. Interestingly, $\mathcal{H}_{10}^{S}$ outperformed $\mathcal{H}_{5}^{S}$ on both $C_{te}^{s}$ and $C_{hv}^{s}$, indicating the relative importance of feature targets within RIDE-Net compared to re-optimization.

Significantly lower variability in c-index can be observed for $\mathcal{H}_{10}^{S}$ compared to $\mathcal{H}^{R18}$. This suggests that RIDE-Net exhibited substantially greater consistency in performance compared to conventional deep learning methods across both $C_{te}^{s}$ and $C_{hv}^{s}$.

When considering feature reproducibility, the ten top features comprising $F^{D}$ had a mean ICC of 0.83 (Fig. 2B) which was markedly improved compared to $F^{R}$ (mean ICC of 0.63). In other words, deep radiomic features extracted via RIDE-Net were markedly more reproducible compared to traditional radiomics features.

Finally, when $\mathcal{H}_{10}^{S}$ was trained with 5%, 10%, or 30% of $C_{tr}^{s}$ in few-shot analysis, it achieved consistent validation c-indices in the range of 0.65-0.66, which significantly surpassed the standard ResNet with c-indices of 0.55-0.62 (Fig. 2C). RIDE-Net thus demonstrated significantly higher predictive accuracy than its counterparts while requiring only a small fraction of the training data, underscoring how the domain knowledge encoded in RIDE-Net during pre-training enables reliable downstream task prediction even with limited data.

## V. Concluding Remarks

In this study, we presented Radiomics Initialized Deep Embedding Network (RIDE-Net), a truly integrated radiomics-deep learning model which exploits radiomic descriptors as a pre-training objective for a residual learning network. Comprehensive evaluation of RIDE-Net demonstrated that it can be refined for accurate, consistent, and reproducible downstream predictions with a fraction of the data needed for other methods, while maintaining feature reproducibility and not suffering significant model variability. We demonstrated the predictive power of a RIDE-Net-based model in prognosticating overall survival in renal cancers relative to baseline radiomics and DL methods, using a publicly available multi-institutional cohort of CT scans. Future applications of RIDE-Net will include applications to other diseases and other imaging modalities. We will also consider testing alternative feature selection methods for pre-training feature targets to fully understand how to best encode disease related information within a neural network.

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
