# OpenReview forum: "Radiomics Initialized Deep Embedding Network (RIDE-Net) to Prognosticate Survival in Renal Cancers"
_IEEE.org/EMBS/BHI/2025/Conference — BHI 2025_

### Official Review · Reviewer_ek9Q · 2025-07-18
**Innovative hybrid design to solve an important problem**

**Confidence:** 4
**Clarity Of Writing:** fair
**Clinical Significance:** good
**Methodological Novelty:** good
**Overall Rating:** 5
**Final Rating:** 6

**Experiments And Results:**

fair

**Questions For The Authors:**

N/A

**Strengths:**

Intuitive two-stage training pipeline, addressing limitations of prior feature- and decision-level fusion methods.

Significantly outperforms baseline models (standard radiomics and ResNet) in survival prediction tasks, as shown by higher c-indices and lower variance across trials.

**Summary Of The Paper:**

This paper introduces RIDE-Net, a deep learning framework that integrates radiomic feature supervision into the pretraining of a 3D ResNet for survival prediction in renal cancers. The model is first trained to predict selected radiomic features (associated with survival outcomes), then fine-tuned for direct survival risk prediction.

**Weaknesses:**

broader applicability across modalities or disease sites is not explored.

 Performance is sensitive to the choice and number of radiomic features used in pretraining (as shown by drop in c-index from HS₁₀ to HS₅), which may require careful tuning for new tasks.

The paper briefly mentions using LASSO for radiomic feature selection but does not deeply analyze which features were most influential or how domain knowledge shaped those choices.


While the model performs well, no discussion is provided on computational complexity or training time, which could be a concern in large-scale clinical deployment.

---

### Official Review · Reviewer_VFPt · 2025-07-18
**Radiomics Initialized Deep Embedding Network: A Promising Approach to Enhancing CNNs with Radiomics Descriptors**

**Confidence:** 4
**Clarity Of Writing:** good
**Clinical Significance:** great
**Methodological Novelty:** good
**Overall Rating:** 7
**Final Rating:** 7

**Experiments And Results:**

great

**Questions For The Authors:**

The paper would benefit from further clarification on what is being learned with respect to the data distribution, especially regarding how the modes associated with target Y are handled within the framework. A discussion of how these modes fit into the overall model and how the authors address them in terms of distribution would be valuable. Additionally, there is a lack of discussion on the statistical principles and machine learning aspects, such as the model description, number of parameters, complexity, and training costs. Also, how generalizable is the RIDE-Net approach to other cancer types or diseases with different imaging characteristics? Would the feature selection process need to be modified significantly for those cases?

**Strengths:**

The paper proposes to integrating radiomics descriptors with deep learning models (CNNs), which significantly enhances the prediction of survival outcomes in renal cancer. RIDE-Net outperforms traditional models like Cox regression and standard CNN-based approaches in both predictive accuracy and consistency. The method demonstrates high robustness, with reduced variation across training runs, making it suitable for clinical applications. The model is also shown to performs well with limited data. The multi-institutional validation further strengthens the generalizability of the results, ensuring the model's potential for broader application. The paper effectively uses elaborative figures to explain the framework and results, helping in understanding the proposed methodology.

**Summary Of The Paper:**

Radiomics Initialized Deep Embedding Network (RIDE-Net) provides a promising approach to enhance CNN models by leveraging radiomics descriptors. This innovative integration improves model performance in predicting survival outcomes in renal cancers using CT scans.  The authors evaluated the model using multi-institutional data from 510 patients and found that RIDE-Net outperformed traditional methods like Cox regression and standard ResNet models in both predictive accuracy and reproducibility. Overall, the paper is well-written and presents a detailed study of the proposed methodology. However, some improvements are suggested to enhance its clarity and make it more readable to a wider audience.

**Weaknesses:**

The step 1,2,3 in method II may help with more clarity. For example, in step1, it is not clear what input is given and what terminology is used for it corresponding to F_i^R and  F_i^D. Similarly, Step 2 does not specify whether the pretrained weights are fixed, with only the fully connected layer being trained, or if the pretrained weights are used solely for initialization. While Figure 1 illustrates the framework well, including the mathematical notations in the figure would improve clarity and consistency in terminology. Additionally, a more detailed description of Figure 1 would be helpful, particularly by separately explaining parts A, B, and C, similar to how Figure 2 is described. The paper also lacks discussion on statistical and ML aspects, computational cost and time associated with the RIDE-Net model.

---

### Official Review · Reviewer_kXgP · 2025-07-20
**The paper makes a compelling case for radiomics-guided pre-training of CNNs through the RIDE-Net framework. By embedding radiomic descriptors into a 3D ResNet-18 and then fine-tuning on survival outcomes, the authors achieved significantly improved c-index and lower model variability than existing models. The methodological design is both intellectually rigorous and clinically applicable.**

**Confidence:** 5
**Clarity Of Writing:** great
**Clinical Significance:** excellent
**Methodological Novelty:** excellent
**Overall Rating:** 8

**Experiments And Results:**

excellent

**Questions For The Authors:**

1. Can the RIDE-Net framework generalize to **multi-modal input** (e.g., PET/CT, MRI)?
2. Is there a plan to incorporate **explainability tools** (e.g., Grad-CAM, SHAP)?
3. Could transfer learning be applied from renal cancer to other tumor types?
4. Are the radiomic features chosen stable across different segmentation tools or protocols?
5. How might different image pre-processing (e.g., normalization strategies) impact RIDE-Net’s performance?

**Strengths:**

* Truly integrated hybrid architecture: radiomics + DL
* Clinically validated on diverse, multi-center data
* Significantly better performance (c-index) than baseline models
* Low variance across 50 random initializations (robustness)
* High reproducibility of deep features (ICC = 0.83 vs 0.63)
* Effective in **few-shot settings** (even with 5% of data)
* Thoughtful evaluation design (ablation, reproducibility, variability)

**Summary Of The Paper:**

* **Objective**: To create a hybrid deep learning model (RIDE-Net) for accurate and reproducible survival prediction in renal cancer using CT images.
* **Methodology**:

  * Step 1: Select radiomics features related to survival using LASSO on CT data.
  * Step 2: Pre-train a deep CNN to predict these radiomics features (radiomics embedding).
  * Step 3: Fine-tune the network to predict patient survival (Cox loss).
* **Datasets**:

  * 5195 unlabeled scans for pre-training
  * 510 labeled scans from 10 institutions for survival modeling
* **Results**:

  * RIDE-Net c-index: **0.67 (test), 0.66 (validation)**
  * Outperformed ResNet-18 (0.60, 0.62) and radiomics-only Cox model (0.56, 0.62)
  * Achieved high **intra-class correlation (ICC = 0.83)** for reproducibility
  * Retained high accuracy in few-shot settings (with 5%–30% data)

**Weaknesses:**

* No inclusion of **other modalities** (e.g., MRI or histopathology)
* Lack of **explainability** – clinical interpretability of deep features remains unclear
* Radiomics feature selection still depends on handcrafted operators (potential bias)
* May not generalize outside of renal cancer without extensive revalidation
* Hyperparameters (e.g., number of selected radiomics features M) influence performance significantly

---

### Official Review · Reviewer_zVjM · 2025-07-20
**Review of RIDE-Net**

**Confidence:** 3
**Clarity Of Writing:** great
**Clinical Significance:** fair
**Methodological Novelty:** good
**Overall Rating:** 7

**Experiments And Results:**

good

**Questions For The Authors:**

How about the performance of survival analysis without the pretraining process?

**Strengths:**

This paper employs a pre-training strategy, enabling the model to leverage large-scale unlabeled data, which significantly enhances its generalizability. Through fine-tuning on survival datasets, the model further improves its prognostic prediction performance, achieving robust results.

**Summary Of The Paper:**

Popular machine learning approaches in medical imaging offer complementary benefits for disease analysis. Radiomic features provide interpretable, quantitative measures linked to disease traits, while CNNs extract complex patterns robust to imaging variations. To combine these strengths, authors propose Radiomics-Initialized Deep Embedding Network (RIDE-Net), which integrates radiomic features into a CNN framework.

**Weaknesses:**

The lenght of the abstract may be too long for a paper with only 6 pages.